# Wastewater Surveillance of SARS-CoV-2: A Comparison of Two Concentration Methods

**DOI:** 10.3390/v16091398

**Published:** 2024-08-31

**Authors:** Christina Diamanti, Lambros Nousis, Petros Bozidis, Michalis Koureas, Maria Kyritsi, George Markozannes, Nikolaos Simantiris, Eirini Panteli, Anastasia Koutsolioutsou, Konstantinos Tsilidis, Christos Hadjichristodoulou, Alexandra Koutsotoli, Eirini Christaki, Dimitrios Alivertis, Aristides Bartzokas, Konstantina Gartzonika, Chrysostomos Dovas, Evangelia Ntzani

**Affiliations:** 1Department of Hygiene and Epidemiology, Faculty of Medicine, University of Ioannina, 45100 Ioannina, Greece; ch.diamanti@uoi.gr (C.D.); lnousis@hotmail.com (L.N.); g.markozannes@uoi.gr (G.M.); nsimantiris@ionio.gr (N.S.); eir_panteli@hotmail.gr (E.P.); ktsilidi@uoi.gr (K.T.); akoutsot@uoi.gr (A.K.); 2Department of Microbiology, Faculty of Medicine, School of Health Sciences, University of Ioannina, 45110 Ioannina, Greece; pbozidis@uoi.gr (P.B.); kgartzon@uoi.gr (K.G.); 3Laboratory of Hygiene and Epidemiology, Faculty of Medicine, University of Thessaly, 22 Papakyriazi Str., 41222 Larissa, Greece; mkoureas@med.uth.gr (M.K.); mkiritsi@uth.gr (M.K.); xhatzi@med.uth.gr (C.H.); 4National Public Health Organization, 15123 Athens, Greece; a.koutsolioutsou@eody.gov.gr; 51st Division of Internal Medicine & Infectious Diseases Unit, University Hospital of Ioannina, Faculty of Medicine, University of Ioannina, 45500 Ioannina, Greece; eirini.christaki@uoi.gr; 6Department of Biological Applications and Technology, University of Ioannina, Ioannina 45110, Greece; aliverti@uoi.gr; 7Laboratory of Meteorology, Department of Physics, University of Ioannina, Ioannina, Greece; abartzok@uoi.gr; 8Diagnostic Laboratory, School of Veterinary Medicine, Faculty of Health Sciences, Aristotle University of Thessaloniki, 54627 Thessaloniki, Greece; dovas@vet.auth.gr; 9Center for Evidence Synthesis in Health, Department of Health Services, Policy and Practice, School of Public Health, Brown University, Providence, RI 02912, USA; 10Biomedical Research Institute, Foundation for Research and Technology, 45110 Ioannina, Greece

**Keywords:** SARS-CoV-2, wastewater surveillance, polyethylene glycol, skimmed milk flocculation

## Abstract

Wastewater surveillance is crucial for the epidemiological monitoring of SARS-CoV-2. Various concentration techniques, such as skimmed milk flocculation (SMF) and polyethylene glycol (PEG) precipitation, are employed to isolate the virus effectively. This study aims to compare these two methods and determine the one with the superior recovery rates. From February to December 2021, 24-h wastewater samples were collected from the Ioannina Wastewater Treatment Plant’s inlet and processed using both techniques. Subsequent viral genome isolation and a real-time RT-qPCR detection of SARS-CoV-2 were performed. The quantitative analysis demonstrated a higher detection sensitivity with a PEG-based concentration than SMF. Moreover, when the samples were positive by both methods, PEG consistently yielded higher viral loads. These findings underscore the need for further research into concentration methodologies and the development of precise protocols to enhance epidemiological surveillance through wastewater analysis.

## 1. Introduction

Following the emergence of SARS-CoV-2 and the subsequent declaration of a pandemic by the World Health Organization (WHO) in March 2020, [1] the scientific community turned to Wastewater-Based Epidemiology (WBE). Considering that the SARS-CoV-2 viral genome had been detected in the feces of not only symptomatic but also asymptomatic patients [2,3,4,5,6,7] and that the successful detection and quantification of the viral genome in the feces from patients and carriers was feasible, wastewater surveillance could provide a complementary picture of the virus dispersal in the community [8,9]. By quantitatively measuring the viral load, virus monitoring using wastewater is independent of the level of clinical surveillance in the population and allows the calculation of the geographic and temporal spread of the virus in a community. These data, together with clinical surveillance, can support local decision-making, particularly in regions where clinical surveillance may not be as robust.

In September 2020, the US Centers for Disease Control and Prevention (CDC), established the National Wastewater Surveillance System (NWSS) with the goal of continuously monitoring the course of the SARS-CoV-2 pandemic through wastewater to enable the early mobilization of authorities to prevent the spread of SARS-CoV-2 [10]. The EU followed in due course in using wastewater-based surveillance to identify potential dangers deriving from emerging diseases of high public health concern, according to the communication from the Health Emergency Preparedness and Response Authority (HERA) that was published in September 2021 [11]. The monitoring of wastewater is currently assisting EU member states to keep an eye on COVID-19 and its variations. Currently, 1370 wastewater treatment facilities are routinely examined throughout the EU. These recommendations were quickly implemented, assisting in the early detection of the virus and its mutations throughout the EU [12].

Since the beginning of the COVID-19 pandemic, research teams around the world have been searching for the most effective and reliable wastewater monitoring protocol for the epidemiological surveillance of the SARS-CoV-2 virus [13,14,15,16]. The difficulty in handling wastewater samples due to the complexity of the substances and inhibitors they contain raises concerns about the accurate detection of pathogenic viruses by an RT-PCR. According to the CDC recommendations, the sample concentration is the most critical step in identifying the viral genome of SARS-CoV-2 [17]. Numerous concentration techniques have been investigated, including ultrafiltration [8,18], ultracentrifugation [19], polyethylene glycol 8000 precipitation (PEG) [20,21], aluminum flocculation [22], skimmed milk flocculation (SMF) [23] and electronegative membrane filtration [24,25]. Among the protocols evaluated in these settings, the attributes of cost and convenience were also considered, as well as the current capacity status of the community where the surveillance takes place.

Both PEG precipitation and SMF are capable of efficiently concentrating viruses from large water volumes, making them suitable for environmental surveillance [13] and they are cost-effective methods, using relatively inexpensive reagents and materials [26,27]. The accessibility of equipment is also a crucial factor, especially for studies conducted in basic or resource-limited laboratories. PEG precipitation and SMF do not necessitate the use of advanced or specialized instruments, which aligns with the capacity of many laboratories [8,19,23,28,29]. The practicality of the methods in basic laboratory settings is another important criterion. Methods, such as PEG precipitation and SMF, that are straightforward and can be implemented without extensive technical expertise or infrastructure are preferred. The aim of this present study was to compare these two concentration methods, skimmed milk flocculation and PEG precipitation, to evaluate their performance in SARS-CoV-2 detection.

## 2. Materials and Methods

### 2.1. Sampling Strategy and Procedure

Forty samples were collected from the input of the wastewater treatment plant (WWTP) of the city of Ioannina, Greece, during the months of June to December 2021, with a sampling frequency ranging from 1 to 3 times per week. The municipal WWTP, which serves a population of 120,000 people, collects household and industrial wastewater, as well as rainwater [30,31]. The wastewater samples (1 L) were subsampled from 24-h composites of the total WWTP influent collected by the WWTP personnel using a WaterSam WS Porti 1 portable water sampler. The samples were placed in an isothermal container with ice and transported within 2 h to the Laboratory of Hygiene and Epidemiology (University of Ioannina, Ioannina, Greece), where they were stored at 4 °C for a maximum of 24 h until they were processed further.

### 2.2. Wastewater Concentration

All the samples were analyzed using both techniques, PEG precipitation (with NaCl) and SMF. These methods were chosen because they can handle large volumes of water for the concentration, are economical, do not require specialized laboratory equipment, and can be supported in a basic laboratory.

PEG precipitation has been widely used to concentrate viruses from water matrices [32,33,34]. In this study, we relied on the research of Mull, Hill, Wu et al., and modified the protocol as follows: 100 mL of the collected wastewater sample was stirred with 2 g of glycine for 30 min. The sample was then divided into two 50 mL centrifuge tubes (Falcon tubes) and centrifuged at 4700× *g* for 30 min at 4 °C to remove the large particles. The supernatant from each Falcon tube was transferred to a new centrifuge tube, containing 5.25 ± 0.1 g of PEG 8000 and 1.182 ± 0.1 g of NaCl and mixed well until the sample was homogenized. The sample was then centrifuged at 12,000× *g* for 2 h at 4 °C. At the end of the centrifugation, the supernatant was discarded, and the Falcons were placed back in the centrifuge for a short new centrifugation of 5 min under the same conditions (12,000× *g*, 4 °C). The supernatant was discarded again. The pellet was then resuspended separately in each centrifuge tube with 400 µL of RNAse-free water. Finally, after moderate vortexing and mild centrifugation for a brief period (3–5 s), the resuspended pellet from each sample (from both Falcons) was collected in 1.5 mL Eppendorf tubes.

The SMF method is based on the absorption of the viruses into the skimmed milk flocks. In this work, we followed the concentration method described by Calgua B. et al. (2008) [35]. The treatment process started with the preparation of artificial seawater (ASW). Then, 5 g of sea salt was added to 100 mL of double distilled water (ddH_2_O), and the solution was shaken until it became transparent. Then, the pre-coated skimmed milk (SM) solution (1% *w*/*v*) was prepared by adding 5 g of powdered skimmed milk to 100 mL of ASW. The pH of the solution was checked and stabilized at a value of 3.5 with HCl 1 N. The pH of the wastewater samples was adjusted in the same way to 3.5. From 100 mL of this solution (ASW and SM), 10 mL was added to each of the previously acidified 1-L solution samples. The samples were stirred for 8 h at 25 °C and left to stand for a further 8 h to allow the milk “flakes”, which may have adsorbed virus molecules, to settle by gravity. After 16 h, the excess was carefully removed with a vacuum pump to avoid disturbing the sediment. The final volume containing the precipitate was transferred to the centrifuge tubes (Falcon 50 mL) and centrifuged at 6,000 rpm for 40 min at 12 °C. After the centrifugation, the supernatant was carefully discarded and the precipitate was reconstituted with 1mL of phosphate buffer 0.2 M at a pH of 7.5 (1:2 *v*/*v* from a solution of Na_2_HPO_4_ 0.2 M and NaH_2_PO_4_ 0.2 M) before being further transferred to a new Falcon tube (15 mL). This process was continued in the remainder of the Falcon tube (50 mL) containing the sediment. Finally, the phosphate buffer was added until the final volume was 10 mL.

### 2.3. Physicochemical Parameters

The physical–chemical parameter analysis was performed on the same day that the samples were received in the laboratory by standard laboratory equipment, according to the American Public Health Association methods [36]. These parameters were pH (APHA 4500-H + B), electrical conductivity (APHA 2520 B), Chemical Oxygen Demand (COD) (APHA 5220 D), Biochemical Oxygen Demand (BOD) (APHA 5210 B), total phosphorus (Ptot) (APHA 4500-P E), and ammonium nitrogen (NH_4_N) (APHA 4500-NH3 C).

### 2.4. RNA Extraction and RT-qPCR Analysis

The Macherey-Nagel Viral RNA and DNA isolation kit from NucleoSpin Microbial DNA was used to isolate the viral genome and the manufacturer’s exact instructions were followed, resulting in 35 µL of the final volume of the isolated nucleic acid solution. The molecular detection of SARS-CoV-2 was performed using a commercially available RT-qPCR assay (Viasure SARS-CoV-2 Real-Time PCR Detection Kit). The assay was based on the amplification of a conserved region of the *ORF1ab* and *N genes* for SARS-CoV-2 using specific primers and fluorescent labeled probes.

A total of 5 μL of the RNA extract was transferred to the reaction tubes containing 15 μL of PCR reagents. The RT was performed at 45 °C for 15 min, and the amplification was performed for 1 cycle of 95 °C for 2 min and 45 cycles consisting of 95 °C for 10 s and 60 °C for 50 s, followed by a final cooling step at 4 °C. The RT-qPCR reactions were performed in triplicate and both the negative control (grade water) and the positive control (according to the kit instructions) were included. The fluorogenic data were collected during the extension step through the FAM (*ORF1ab* gene), ROX (*N gene*), and VIC (the extraction control). A CFX-96 instrument (Bio-Rad Laboratories) was used for all the amplification reactions. The efficiencies of both the RNA extraction and RT-qPCR were assessed in all the samples using the extraction control (EC), which was added during the lysis step in the viral genome isolation.

### 2.5. Quantification

The viral load quantification was based on a standard curve constructed for the *N gene* and expressed as genome copies/L of wastewater. Twist Synthetic RNA Control 2 (Twist Bioscience) is one of the initial isolates of SARS-CoV-2 and served as a reference sequence including the whole genome of Wuhan. It was used to construct the standard curve with serial 10-fold dilutions in triplicate. The efficiency was determined based on the equation of efficiency: (E = (−1+10^(−1/slope)^) × 100 [37]. The values for our curve are defined as follows: e% = 99.64, slope  = −3.353, y intercept = 38.102, and Rsq = 0.999. The number of Ct (the cycle threshold) was used to determine the positivity/negativity of the samples. As a starting point, a Ct value of 43 was used as a positivity threshold. The samples that gave a value above the positivity threshold (Ct > 43) were considered negative for SARS-CoV-2. Amplification with a Ct value less than 43 had to occur in two out of three replicates for a sample to be considered positive for SARS-CoV-2. The sample was considered equivocal and recorded as negative if only one of the replicates showed amplification below 43.

### 2.6. Statistical Analysis

The sample characteristics were descriptively reported using the mean ± SD, median and interquartile range (IQR), or frequencies (percentages) as appropriate. The Mann–Whitney test for the continuous data and Fisher’s exact test for the categorical data were used to assess the differences in the sample characteristics.

The diagnostic performance metrics (sensitivity, specificity, positive predictive value (PPV), and negative predictive value (NPV)) for the two concentration methods were calculated, and the comparisons were made using the McNemar test.

The statistical significance was considered as *p* < 0.05. All the statistical analyses were performed using Microsoft Excel/SPSS.

## 3. Results

Table 1 lists the mean and median values and the minimum and maximum values for each physicochemical parameter assessed. Across the 40 analyzed samples, the minimum pH measured was 7.00 and the maximum was 8.26. The averages of the COD and BOD were 360.02 mg/L and 140.33 mg/L, respectively, and the median values for Ptot and NH4N were 11.73 mg/L and 10.53 mg/L, respectively. The range of electrical conductivity (EC) was 577–1397 mS/cm. As they pertained to samples from an extended time series and various environmental circumstances over a long period of time (June–December), the observed variation in the range of values for the physicochemical parameters was expected.

Using the positivity Ct threshold of 43 cycles and PEG as the gold standard, we observed a 50% negativity (*n* = 20) when the samples were concentrated by the SMF method, while no negative sample was detected by PEG (Appendix A). Considering a positivity threshold of 40 (Ct = 40), the sensitivity of the concentration by the SMF method reached 47%, while the specificity was 100%. Accordingly, by changing the positivity threshold to 45 (Ct = 45), the sensitivity for SMF remained small (53%). The SMF positive predictive value was 100% regardless of the Ct threshold. On the other hand, very low values occurred for the SMF negative predictive value, suggesting that the chance that the negative samples were indeed negative was quite low (NPV40 = 17%, NPV45 = 10%), considering this technique inferior to PEG. Finally, the accuracy of the SMF method was low at both Ct thresholds (a small difference of 2%) (Appendix A).

During the quantification, the positive samples resulting from both concentration techniques were evaluated, with the aim of determining the virus load in each sample (Figure 1). From the box plot, the median of the PEG method is closer to the middle of the box, indicating a more balanced distribution, while the position of the median of the SMF method is closer to the bottom of the box, indicating a skew toward lower values.

Furthermore, in the SMF method, the whiskers of the box plot are smaller and placed closer to the box, which indicates the smaller range of the method. In contrast, according to the axes of the second box plot, the PEG method represents a dataset with a wider distribution of values, indicating a larger range of values. These findings revealed a significant difference between the results produced by the two methods, with a strong advantage leaning towards the use of the PEG method. This contrast in results can also be seen in Appendix A.

The results of the two concentration methods show a statistically significant difference in the observed viral loads with a particularly higher virus recovery using PEG (Figure 2) (*p* < 0.05).

## 4. Discussion

In this study, we compared two concentration protocols based on PEG precipitation and SMF, respectively, for their performance in the environmental surveillance of SARS-CoV-2 in an urban setting. The two protocols were chosen because they do not require extensive laboratory resources and expensive laboratory consumables, also allowing seamless wastewater monitoring in resource-poor environments. We observed an important advantage of the PEG protocol over the SMF protocol. The PEG protocol outperformed the SMF protocol in terms of the sample concentration time, detection sensitivity, and virus recovery. Specifically, the PEG protocol was faster, because the samples were concentrated on the same day, rather than overnight as in the SMF protocol. There was also a significant difference in the detection sensitivity between the two methods, as many samples that were considered negative by the SMF concentration tested positive by the PEG precipitation. This difference remained for the samples that were positive by both methods, with the PEG protocol consistently measuring higher viral loads indicating higher recovery. These findings were further supported by the quantification of the results using an RT-qPCR, which demonstrated that the PEG concentration method had a much higher detection sensitivity compared to that of the SMF method. However, there were three samples where the SMF method had marginally better detection than the PEG (Appendix A). The small difference in performance between the SMF and PEG methods for these samples could be attributed to variability in the composition of the effluent during the collection period. Wastewater is a highly complex and dynamic matrix, and its composition can fluctuate due to many factors, including environmental conditions, industrial discharges, and population behavior. This variability can affect the efficiency of virus collection methods, potentially making one method more effective under certain conditions [38]. The hypothesis that the selection of the best concentration technique could significantly affect the accuracy and reliability of virus detection and quantification is reinforced by this obvious variation in detection sensitivity. Furthermore, to correlate (using Pearson’s correlation) the physicochemical parameters measured and the presence of the virus, there was no statistically significant result confirming a dependent relationship (Appendix A).

This work is well integrated with the published literature on the environmental surveillance protocols for SARS-CoV-2. According to Kumar et al., PEG precipitation was the most widely used concentration technique due to its selectivity and resilience to PCR inhibitors [39]. SMF has been shown to be an effective and inexpensive method for virus concentration in a variety of water samples, including rivers [40], seas [35], groundwater [41] and wastewater [42]. It is also suitable for resource-poor areas, as it does not require expensive equipment, and the entire technique can be carried out in one day. Despite the advantages of the skimmed milk flocculation procedure, the nature of the wastewater sample can cause problems, as wastewater samples are unstable and difficult to handle [23]. Similar problems appear to have occurred during the concentration step in this study, resulting in the advantage of PEG over SMF.

In the study of Salvo et al. (2021) [16], three different concentration methods in wastewater samples were investigated, including PEG precipitation and SMF. The surrogates of enveloped and non-enveloped viruses were used, such as SARS-CoV-2 (an enveloped virus). Their findings revealed that PEG precipitation was more effective for viral concentration for both virus types, as it was more sensitive than SMF. The PEG method also proved to be more sensitive than SMF, as the sample volume to be tested was 10 times greater for the PEG concentration method (500 mL) than for the SMF (50 mL) [16]. A similar study on wastewater samples in Spain revealed that the recovery rate of PEG was higher compared to the aluminum-based adsorption–precipitation approach [43].

Based on a recent study that compared concentration techniques and identified the most effective one, wastewater samples were infected with Feline calicivirus (FCV), a non-enveloped, positive-sense RNA genome virus [44]. In this study, PEG precipitation was shown to have the highest performance for FCV recovery [14]. PEG precipitation and polyaluminum chloride (PAC) flocculation were two approaches that stood out. After the first method comparison, another test was performed between the two main methods to highlight the one with the best recovery of SARS-CoV-2. The second comparison revealed that PAC flocculation had a lower limit of detection than PEG precipitation [14].

Interestingly, there are also studies that suggest the opposite, i.e., the superiority of SMF over PEG precipitation. One of these studies was conducted in Portugal in 2020 by Philo et al. [23] Comparing the performance of three concentration methods for the recovery of porcine epidemic diarrhea virus (PEDV) (used as a surrogate for SARS-CoV-2), it was shown that PEG had the worst recovery performance while SMF performed slightly better [13]. Similar results were presented in the research by Philo et al., who showed that, when evaluating different concentration techniques, SMF had the highest recovery rate [23]. We have discovered that there are differences between the previously mentioned concentration procedures and the ones used in our research. In all three studies, longer PEG precipitation times (overnight) were utilized, along with greater sample volumes (500–1000 L) and fewer centrifugations, which may be the cause of the different results.

In summary, determining which is the ideal viral concentration method for SARS-CoV-2 is extremely challenging because the comparison setting of the different concentration methods varies widely. There are studies that highlight the volume used when concentrating the sample [16], while others focus on its quality by assessing its physicochemical properties [15]. When comparing the methods, the SARS-CoV-2 virus is often replaced by surrogates from other virus groups, such as non-enveloped and enveloped viruses. There are also many differences in the presentation of the results, with many studies reporting only the recovery and efficiency when using a particular method [45,46] and not the method’s Limit of Quantification (LOQ) or Limit of Detection (LOD), while others report the recovery efficiency and LOD but not the LOQ [22,24,43], while there are also studies that report the LOQ but not the LOD [47], which does not automatically make them comparable across the range of results.

Various methods are used by European countries, as indicated in Table 2. Based on the data available from national dashboards, a wide range of approaches are implemented, each with its own advantages and limitations in terms of cost, efficiency, and the volume of inputs.

For example, PEG—NaCl-based precipitation appears to be widely used in Central and Eastern European countries as it usually requires less equipment, making it accessible for wider application in areas with moderate resources [48,50,58]. Ultrafiltration is widespread in Western and Northern European countries, reflecting the ability of these countries to invest in more sophisticated equipment. This method is highly effective, although its high cost may limit its application in less-resourced settings [8,49,51,52,53,55,56,57,59,60]. The direct capture method is highly effective and is used in countries with a robust infrastructure. However, its high cost reflects the need for expensive and advanced equipment, making it less accessible in lower income areas [54,62].

Therefore, there seems to be a clear separation in the choice of methods based on the economic capacity of the countries. Higher income states tend to use ultrafiltration or direct capture methods, which, while more expensive, provide fast and reliable results. In contrast, low-cost methods such as PEG-based precipitation are still effective but may have limitations in terms of speed and efficiency.

In addition, the required input volumes vary, with ultrafiltration generally requiring smaller volumes (up to 30 mL in France [53]) compared to precipitation methods (up to 500 mL in the Czech Republic [50]). This variation may affect the choice of method, depending on the sample size availability.

Finally, most methods are designed for same day analysis, which is crucial for timely public health responses. Potentially, there are protocols that exhibit delays (where an overnight step is required) and potentially delays results, although this is effectively unenforced by many countries.

According to the findings of our own investigation, PEG precipitation was more sensitive than SMF for the concentration of the SARS-CoV-2 virus in municipal wastewater samples of the city of Ioannina. This advantage, together with its low cost, automatically makes it suitable for SARS-CoV-2 monitoring in the field of wastewater epidemiology. Equally important is the reference to the originality of our research. The two approaches were compared in parallel using actual sewage samples in a period with many fluctuating virus circulations over a lengthy period of time. PEG performed better throughout the time series that included samples with various SARS-CoV-2 loads (differences of up to 5 orders of magnitude from 0 × 10^0^ to 1.7 × 10^5^ GC/L).

Further investigation and comparison of different protocols at all stages of the sample analysis are needed to determine the most appropriate approach for successful wastewater monitoring. An important objective of optimizing the concentration methods is to more precisely time the appearance of the virus in a community and more accurately monitor its circulation over time. This is crucial for timely public health responses and understanding the dynamics of viral spread. However, it is essential to acknowledge the limitations of our study. The variation in detection sensitivity among different methods, the influence of the sample quality, and the potential for different environmental factors impacting the results highlight the complexity of wastewater-based epidemiology. Addressing these limitations through further research and method refinement is vital for improving the accuracy and reliability of the virus monitoring. Further research is needed on the use of appropriate sampling, storage, pre-treatment, and isolation strategies for the viral genome of SARS-CoV-2 from wastewater samples.

## Figures and Tables

**Figure 1 viruses-16-01398-f001:**
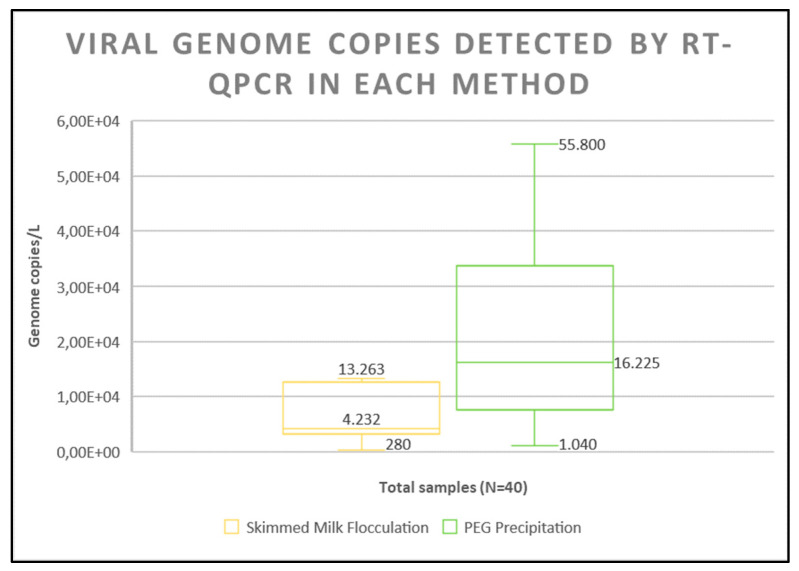
Box plots for genome copies/L according to the two concentration methods.

**Figure 2 viruses-16-01398-f002:**
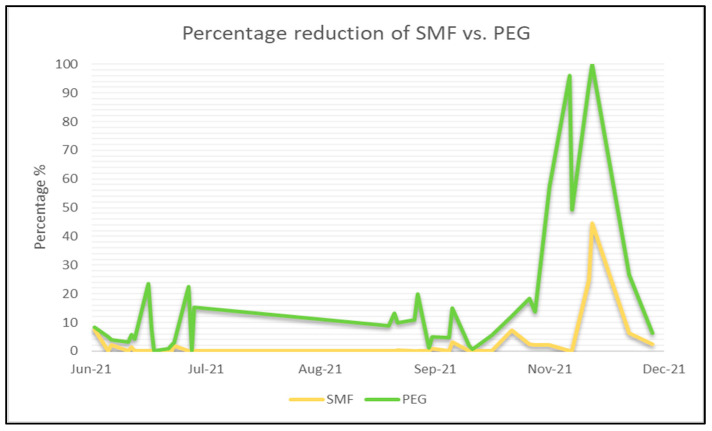
Percentage reduction in the results of the two methods and their comparative differences. Figure 2 presents a time series analysis of the data, illustrating that the proportions of positive results for both methods were consistently concordant over the entire evaluated time frame.

**Table 1 viruses-16-01398-t001:** General characteristics of the samples.

	pH	EC (μS/cm)	COD (mg/L)	BOD (mg/L)	P_tot_ (mg/L)	NH_4_N (mg/L)
MIN	7	577	129	52	7.85	7.45
MAX	8.26	1397	641	213	17.8	12.65
MEAN	7.78	989.60	360.02	140.33	11.71	10.48
MEDIAN	7.78	1007.50	366	138.50	11.73	10.53

**Table 2 viruses-16-01398-t002:** SARS-CoV-2 wastewater concentration methods used in European countries.

Country	Concentration Method	Estimated Cost	Equipment Requirements	Throughput	Input Volume	Reference
Austria	PEG—NaCl-based precipitation	low cost	↓	same day	250 mL	Daleiden B. et al., 2022 [48]
Belgium	Ultrafiltration	high cost	↑	same day	50 mL	Bertels X. et al., 2023 [49]
Czech Republic	PEG—NaCl-based precipitation	low cost	↓	overnight step	500 mL	Sovová K. et al., 2024 [50]
Finland	Ultrafiltration	high cost	↑	same day	70 mL	Tiwari A. et al., 2022 [51]
Netherlands	Ultrafiltration	high cost	↑	same day	100–250 mL	Medema G. et al., 2020 [8]
						Izquierdo-Lara R. et al., 2021 [52]
Italy	PEG—dextran method	low cost	↓	overnight step	250 mL	La Rosa G. et al., 2020 [20]
France	Ultrafiltration	high cost	↑	same day	30 mL	Lazuka A. et al., 2021 [53]
Germany	Direct capture method	high cost	↑	same day	40 mL	Bartel A. et al., 2024 [54]
Hungary	Ultrafiltration	high cost	↑	same day	50 mL	Róka E. et al., 2021 [55]
						Róka E. et al., 2022 [56]
Ireland	Ultrafiltration	high cost	↑	same day	200–250 mL	Reynolds L. et al., 2022 [57]
Latvia	PEG—NaCl-based precipitation	low cost	↓	same day	135 mL	Dejus B. et al., 2023 [58]
Luxembourg	Ultrafiltration	high cost	↑	same day	120 mL	Coronastep | Luxembourg Institute of Science and Technology [59]
Slovakia	Ultracentrifugation	high cost	↑	same day	50 mL	Krivonakova N. et al., 2021 [60]
Spain	Aluminum hydroxide adsorption–precipitation	low cost	↓	same day	200 mL	Protocolos de detección de SARS-CoV-2 en aguas [61]
						Randazzo W. et al., 2020 [22]
Sweden	Direct capture method	high cost	↑	same day	40 mL	Perez-Zabaleta M. et al., 2023 [62]

## Data Availability

The dataset is available on request from the authors.

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
