# Peer review of "Wastewater Surveillance of SARS-CoV-2: A Comparison of Two Concentration Methods"

_viruses, 2024, doi:10.3390/v16091398_

Round 1

Reviewer 1 Report

Comments and Suggestions for Authors

The manuscript of Diamanti and co-workers describe a comparison of two concentration methods for the identification of SARS-CoV-2 from wastewater. The authors identified Skimmed Milk Flocculation (SMF) and Polyethylene Glycol (PEG) precipitation methods as approaches that would best fit those available at a low-resource setting.

The comparison of 40 samples convincingly shows that the PEG was the easier and better method. I appreciated the discussion of prior work that showed the opposite conclusion and the possible reasons for the differences. I found the manuscript to be well written and the data clearly presented. No major issues were detected. With the exception of one minor issues that warrants a little more exploration, this paper is suitable for publication.

Minor issue: In general, the authors data showed superior performance of PEG over SMF for each sample. But there are three (see Table S1) where the SMF method had marginally better detection than PEG. What is the explanation for the difference? Is there any significance to the clustering? As these samples are likely collected in a narrow window of time, could there have been a change in the composition of the wastewater during this time which made the SMF method superior?

Author Response

We sincerely appreciate the time and effort you have invested in reviewing this manuscript. Below, you'll find our detailed responses, along with the revisions and corrections marked in the re-submitted documents using track changes.

1) In general, the authors data showed superior performance of PEG over SMF for each sample. But there are three (see Table S1) where the SMF method had marginally better detection than PEG. What is the explanation for the difference? ) Is there any significance to the clustering? 
The small difference in performance between the SMF and PEG methods for these particular samples could be attributed to variability in the composition of the effluent during the collection period. Wastewater is a highly complex and dynamic matrix, and its composition can fluctuate due to many factors, including environmental conditions, industrial discharges, and population behavior. This variability can affect the efficiency of virus collection methods, potentially making one method more effective under certain conditions. Grouping does not appear to provide any additional information about the outcome of these three samples.
Please see the additions we made to lines 207 -277.

2) As these samples are likely collected in a narrow window of time, could there have been a change in the composition of the wastewater during this time which made the SMF method superior?
We agree with the reviewer that changes in the composition of the wastewater could be a contributing factor. Given the narrow time window of sample collection, it is possible that the wastewater composition varied slightly during this period, favoring the SMF method in those specific samples. However, without detailed compositional analysis of the wastewater during each collection period, we cannot conclusively determine the exact cause.

Reviewer 2 Report

Comments and Suggestions for Authors

See attached Word file.

Author Response

Thank you for your thorough review of our manuscript. We have provided detailed responses below, with the corresponding revisions and corrections clearly marked in the resubmitted files using track changes.

1) Abstract, Line 31-34. Replace with, “Quantitative analysis demonstrated higher detection
sensitivity with PEG-based concentration than SMF. Moreover, when samples were positive by
both methods, PEG consistently yielded higher viral loads.”
Τext changes

2) Introduction, Line 76. “of pathogenic viruses by RT-PCR.”
Τext changes

3) Materials & Methods, Line 99. “Forty samples ….”
Τext changes

4) Materials & Methods, Line 126. Use of RNAse free water seems of little value given the high
concentration of RNAses present in wastewater. Why isn’t addition of RNAse inhibitors a
consideration?
Thank you for your valuable feedback. At the time of conducting the research, our primary focus was on ensuring that all reagents and consumables used were RNase-free to minimize external sources of contamination. We acknowledge that the high concentration of RNases present in wastewater could pose challenges for RNA stability. However, due to the limitations in our knowledge and protocols at the time, we did not incorporate the use of RNase inhibitors.
Since then, our protocols have evolved, and we now include an additional step involving the use of RNase inhibitors to further protect RNA integrity during sample processing. This enhancement is based on our improved understanding of the complexities associated with wastewater samples and the challenges posed by endogenous RNases. While the original experiments were conducted without this step, the results obtained remain valid, and we believe that the addition of RNase inhibitors in future work will further strengthen the reliability of RNA analyzes in similar studies.

5) Results. A Table could be provided to compare other concentration procedure variables, like
cost, equipment requirements, time to completion/throughput, input volume, etc. A survey of
methods currently used by EU member states would be icing on the cake.
Thank you for your suggestion. We've added a table that aggregates various variables such as cost, equipment requirements, time to completion, and input volume, for different wastewater collection methods. This table includes data from several European countries based on the scientific literature, providing a comprehensive overview of the methods used today. Please find the table at pages 8-9 and commenting it out on the lines 334-362.

6) Results, Figure 2. It should be noted in the text that precedes Figure 2 that the data presented
represents a time-series and that the proportions of positive results for both methods were
concordant over the time frame evaluated.
Thank you for your comment regarding the need to clarify the nature of the data presented in Figure 2. We agree that specifying that the data represents a time series and that the proportions of positive results for both methods were concordant over the evaluated time frame will enhance the clarity of our presentation.
Changes to the text. Lines 250-253. 

7)Results, Table 2. Table 2 could be removed as it is redundant with data presented in the text.
Thank you for your comments on Table 2. We understand that the information presented in the table may seem redundant since it is also described in the text. To simplify the manuscript and avoid redundancy, we have removed Table 2 and that necessary details are clearly conveyed in the text.

8) Discussion. The authors should note somewhere that the objective of optimizing concentration methods is to more precisely time the appearance of the virus in a community and more accurately monitor the extent of its circulation over time. Study limitations should also be mentioned.
We agree that it’s important to highlight that improving concentration methods is key to understanding the bigger picture of the study. Also, talking about the study’s limitations helps give a fair view of the results.
Please find the changes to the text, lines 374-382.
